# Targeting Platelet in Atherosclerosis Plaque Formation: Current Knowledge and Future Perspectives

**DOI:** 10.3390/ijms21249760

**Published:** 2020-12-21

**Authors:** Lei Wang, Chaojun Tang

**Affiliations:** 1Cyrus Tang Hematology Center, Cyrus Tang Medical Institute, Soochow University, Suzhou 215123, China; lwangLeiW@stu.suda.edu.cn; 2Collaborative Innovation Center of Hematology of Jiangsu Province, Soochow University, Suzhou 215123, China; 3National Clinical Research Center for Hematologic Diseases, the First Affiliated Hospital of Soochow University, Suzhou 215123, China

**Keywords:** platelet, atherogenesis, disturbed flow, hyperglycemia, hyperlipidemia, inflammation, platelet migration

## Abstract

Besides their role in hemostasis and thrombosis, it has become increasingly clear that platelets are also involved in many other pathological processes of the vascular system, such as atherosclerotic plaque formation. Atherosclerosis is a chronic vascular inflammatory disease, which preferentially develops at sites under disturbed blood flow with low speeds and chaotic directions. Hyperglycemia, hyperlipidemia, and hypertension are all risk factors for atherosclerosis. When the vascular microenvironment changes, platelets can respond quickly to interact with endothelial cells and leukocytes, participating in atherosclerosis. This review discusses the important roles of platelets in the plaque formation under pro-atherogenic factors. Specifically, we discussed the platelet behaviors under disturbed flow, hyperglycemia, and hyperlipidemia conditions. We also summarized the molecular mechanisms involved in vascular inflammation during atherogenesis based on platelet receptors and secretion of inflammatory factors. Finally, we highlighted the studies of platelet migration in atherogenesis. In general, we elaborated an atherogenic role of platelets and the aspects that should be further studied in the future.

## 1. Introduction

Atherosclerosis (AS) is a chronic inflammatory disease induced by multiple factors, involving a complex series of circulating blood cells (e.g., platelets and monocytes) and plasma components (e.g., lipoproteins), which interact with vascular cells and initiate atherosclerosis [1,2,3]. AS lesions often occur at the bifurcation or curvature of the large- and medium-sized arteries (e.g., aorta, carotid arteries), where disturbed flow (d-flow) occurs. In addition, a change of the microenvironment in circulation like hypertension, hyperglycemia, or hyperlipidemia can accelerate the formation of atherosclerosis [4,5,6]. When plaques are formed, changes in hemodynamics will exacerbate the tendency to increase plaque in an arterial stenosis environment, making it unstable and causing rupture. Once the plaque ruptures, platelets-rich blood clots are immediately formed to block blood vessels, and ischemic thrombotic events occur.

Platelets are the key mediator of plaque rupture and atherothrombosis. In recent years, many studies have shown that platelets play an inflammatory role as an immune cell and participate in the development of atherosclerosis [7,8]. When the shear stress of the blood flow changes sharply or the vascular microenvironment alters, the platelets circulating in the blood can quickly perceive these signals and respond. Subsequently, they are activated rapidly during endothelial dysfunction and then they adhere to damaged blood vessels to maintain blood vessel integrity [1,9]. Further, activated platelets recruit immune cells, promote their transmigration across the intima, and accelerate the process of atherosclerosis by the engagement of surface receptors or the release of inflammatory factors [10]. Several reviews about platelets in atherosclerosis are available, each with a different emphasis and perspective [1,3,11,12]. Nording et al. [12] summarized the important role of platelets in atherosclerosis and atherothrombosis, and concluded that antiplatelet therapy is not suitable for primary prevention treatment due to its bleeding risk in the treatment of clinical cardiovascular disease. In this review, we try to focus on those basic studies that are related to how platelets, as inflammatory mediators, respond to atherosclerosis risk factors and regulate atherosclerotic plaque formation. Specifically, we illustrate the important roles of molecules derived from platelets in atherogenesis. Finally, we discuss whether platelet migration is involved in the process. We also raise the unsolved questions of platelets in atherogenesis as well as the highlights and perspectives in future research.

## 2. Platelets and Risk Factors of Atherosclerosis

### 2.1. Disturbed Flow Modulated Platelets in AS

Atherosclerosis preferentially develops at the curved bifurcated sites under d-flow [13], while straight blood vessels exhibit uniform laminar flow, which is essential to maintain the vascular homeostasis, and protect against AS. Endothelial cells (ECs) are located in the inner layer of the vascular wall, which is the sensor of hemodynamic shear stress. Under physiological conditions, endothelial cells undergo unidirectional laminar flow, leading to uniform cell alignment, vasodilation, anti-inflammation, and anticoagulation. In contrast, d-flow stimulates proinflammatory responses, including junction damage, leukocyte recruitment, and coagulation [13,14]. Besides, circulating platelets can also rapidly sense and respond to hemodynamic forces to regulate vascular homeostasis. When subjected to d-flow, inflamed ECs express or expose high levels of adhesion molecules (e.g., P-selectin, intercellular adhesion molecule-1 (ICAM-1), vascular cell adhesion molecule 1 (VCAM-1)) [15] and adhesion proteins (e.g., Von Willebrand factor (vWF), Fibrin) [16,17]. These molecules stimulate platelets to adhere and roll to damaged sites via platelet receptors, such as glycoprotein (GP) Ibα, glycoprotein VI (GPVI), or integrin αIIbβ3 (see Figure 1). Subsequently, the adhered platelets can capture the circulating leukocytes, which is limited to the area where P-selectin is expressed under disturbed flow [18]. Interestingly, Tersteeg et al. [19] showed that adherent and activated platelets under shear will expose long negatively charged membrane strands, called flow-induced protrusions (FLIPRs). FLIPRs can capture circulating monocytes and neutrophils in a P-selectin/PSGL-1-dependent manner, and then promote the formation and activation of platelet-leukocyte microparticle complexes, leading to the progression of inflammatory processes. In addition, FLIPR also promotes the formation of platelet microparticles (PMPs), while it is well known that PMPs play an important role in inducing foam cell formation, and promoting atherosclerosis [20]. Although FLIPR promotes platelet-leukocyte microparticle complexes and the release of platelet microparticles, to date, there is no direct evidence that FLIPR plays a role in atherosclerosis or inflammatory diseases in vivo. In addition, platelet endothelial cell adhesion molecule-1 (PECAM-1) can mediate platelet adhesion to endothelial cells under pathological flow, and a lack of PECAM-1 on either the endothelium or platelets results in reduced platelet adhesion to the endothelium [21]. PECAM-1 is also an important factor in atherosclerosis, as a lack of PECAM-1 in ApoE^−/−^ mice will significantly reduce the lesion size of the aortic arch and aortic sinus [22]. Therefore, the atherogenic effect of PECAM-1 may be attributed to platelet adhesion, and the adhered platelets recruit circulating leukocytes to promote vascular inflammation and atherosclerosis. All in all, there is still no conclusive and direct evidence regarding whether and how platelets participate in the formation of early atherosclerotic plaques under d-flow conditions.

### 2.2. Hyperlipidemia Modulated Platelets in AS

Dyslipidemia is a major risk factor of coronary artery disease and is associated with a poor prognosis, such as the major cardiovascular adverse event (MACE) [23,24,25]. The effects of chronic hyperlipidemia are complex, which causes lipid deposition in atherosclerotic lesions, primary endothelial damage, and increased platelet reactivity. Thus, the mechanism leading to enhanced platelet reactivity is one of the key points in the treatment of vascular diseases. Evidence suggests that lipoprotein–platelet interaction plays an important role in atherogenesis. Platelets in hyperlipidemia show high activity and are more prone to be activated and aggregated [8]. In addition to lipid-lowering effects, statins can also effectively reduce the platelet activity [26,27]. Furthermore, platelet can also mediate lipoprotein transport to promote foam cell formation and exacerbate atherosclerosis (see Figure 2).

It is well known that platelets, through specific binding receptors, affect low-density lipoprotein (LDL) per se triggers platelet activation, and enhances platelet aggregation and secretion, whereas HDL desensitizes platelets, underlining the anti-atherosclerotic properties of high-density lipoprotein (HDL) [28,29]. Omics analysis showed that resting human platelets contained 1500 LDL-binding sites and 3200 HDL-binding sites [30]. Human platelets express LDL receptor-related protein 8 (LRP8) [31]. LDL binding to LDL receptors on platelets promotes lipid exchange between LDL particles and platelet plasma membrane, initiates phosphorylation of focal adhesion kinase (FAK), enhances the binding of fibrinogen and integrin αIIbβ3, and induces an increase in intracellular Ca^2+^. The changes of the signal transduction cascade in platelets result in increased sensitivity of platelet-activating agents [32]. Native HDL regulates platelet signaling pathways by binding to platelet HDL receptors (such as SR-BI and apoER2′), as well as by balancing the cholesterol content in platelets to prevent platelet hyperresponsiveness. [33].

The increased lipid in the plasma of patients with hyperlipidemia invades the intimal endothelium in the form of LDL through their receptor, and is oxidized to oxidized low-density lipoprotein (oxLDL) [34]. OxLDL loses its ability to bind to natural LDL receptors, and can be recognized by scavenger receptors (CD36, LOX-1) on platelets to exert additional pro-atherosclerotic effects [8]. OxLDL can enhance the adhesion of platelets to collagen and activated ECs and the formation of platelet-monocyte aggregates [35]. OxLDL binding to CD36 actives platelet and increases exposure of platelet P-selectin and activates integrin αIIbβ3, which may involve the signal pathway including Src family kinases, Syk, and phospholipase C-γ [36,37]. Furthermore, activated platelets internalize oxLDL, while platelets loaded with oxLDL further activate the endothelium, release chemokines to recruit monocytes, and promote foam cell development [38]. CD36 deficiency in ApoE^−/−^ mice can inhibit neointimal hyperplasia and vascular smooth muscle cells (VSMCs) proliferation, which may prevent atherosclerosis and restenosis [39]. Lectin-like oxidized LDL receptor 1 (LOX-1) was identified as another major receptor of oxLDL in human platelets [40]. Activated LOX-1 stimulates adhesion molecules, pro-inflammatory factors, and angiogenic factors in vascular ECs and macrophages [41]. LOX-1 activation can also indicate the occurrence of acute cardiovascular disease after plaque rupture [42]. These studies have shown that oxLDL and its receptors (CD36 and LOX-1) are involved in the regulation of atherosclerotic process. However, there is no direct evidence that specific knockout of CD36 or LOX-1 on platelets in atherosclerosis-prone mice affects atherosclerotic plaque formation.

### 2.3. Hyperglycemia-Modulated Platelets in AS

It is believed that hyperglycemia accelerates the progression of atherosclerosis [43]. Hyperglycemia from non-diabetic patients is also related to the increase in mortality and morbidity of patients with acute stroke [44]. Diabetes can alter the functional properties of many cell types, including ECs and platelets. Many earlier studies have shown that in patients with diabetes mellitus (DM), especially type 2 diabetes mellitus (T2DM), platelet degranulation and cytoplasmic calcification and thromboxane A2 content increase [45]. Hyperglycemia will also promote the binding of platelets and fibrinogen, and enhance p-selectin exposure [46]. Furthermore, platelets from DM patients are less sensitive to natural anticoagulants, such as prostacyclin (PGI2) [47]. These may be the key triggers leading to platelet hyperreactivity and a hypercoagulable state in blood.

Several mechanisms have been proposed to influence platelet activation and platelet hyperreactivity induced by hyperglycemia. Hyperglycemia can cause an increase in non-enzymatic glycosylated LDL (glycLDL), which may lead to changes in protein structure and conformation, as well as changes in membrane lipid dynamics [48]. GlycLDL may increase the expression of platelet key ligand receptors, such as P-selectin and GPIIb/IIIa, by elevating intracellular Ca^2+^ concentration and platelet NO production, and inhibiting changes in platelet membrane dynamics, leading to platelet dysfunction [49]. Recent research showed that chronic hyperglycemia leads to enhanced activation of biomechanical αIIbβ3, which leads to increased shear and erythrocyte dependence of discoid platelet adhesion and aggregation [50]. Moreover, the content of sorbitol in platelets is increased after prolonged high glucose treatment, and the accumulation of sorbitol is closely related to cell swelling. Addition of sorbitol can increase the mean platelet volume (MPV) by regulating platelet microtubule polymerization, and accelerate the production of platelet-related thrombin [51].

In further, circulating platelet-leukocyte aggregates are significantly increased in diabetes patients [52]. Elevated monocyte-platelet aggregates are normally used as an early marker of T2DM [53]. Platelet–leukocyte cross-talk is generally considered to be related to platelet hyper-responsiveness, which may contribute to excess microvascular risk in DM patients. Kraakman et al.’s research found that hyperglycemia triggers the release of S100 calcium-binding protein A8/A9. These proteins then bind to their receptors on Kupffer cells for advanced glycation end products, triggering the production of IL-6, and causing hepatocytes to secrete TPO, which further leads to increased megakaryocyte proliferation and reticulated platelet production [54,55]. Another study reported that hyperglycemia can enhance sodium arsenite-induced megakaryocyte adhesion, platelet P-selectin expression, and leukocyte-platelet aggregation [56]. Megakaryocyte adhesion is often considered one of the markers of atherothrombosis risk [57]. In summary, hyperglycemia can regulate the atherosclerotic process by regulating platelet signaling and the state of platelet receptors.

## 3. Platelet Receptors in Atherogenesis

There are abundant receptors on the surface that bind to extracellular matrix and adhesion proteins to cause platelet adhesion and activation. Major platelet receptors include integrins (αIIbβ3, αVβ3), immunoglobulin superfamily adhesion receptors (GPVI, FcγRIIa), leucine-rich adhesive receptors (GPIb-IX-V complex, toll-like receptors), G-protein-coupled receptors (PAR1, PAR4, P2Y_12_), and C-type lectin receptors (P-selectin) etc. [58,59]. These receptors interact with various adhesion matrices and cells (endothelial, monocytes, smooth muscle cells, etc.) to participate in inflammatory reactions (Figure 3).

### 3.1. Integrin αIIbβ3

The platelet cascade activation is inseparable from the expression of αIIbβ3. The αIIbβ3 on resting platelets is in an inactive state, but when the signal transduction inside the platelets is activated, αIIbβ3 receptor activation and fibrinogen binding further promote platelet aggregation [60]. Numerous clinical trials have shown that the inhibitors of platelet αIIbβ3 receptors can effectively prevent and treat unstable angina pectoris or myocardial infarction caused by acute ischemic events [61]. However, whether platelet αIIbβ3 promotes the development of atherosclerotic lesions is currently controversial. It is reported that the absence of integrin β3 in atherosclerosis-prone mice (ApoE-null and LDLR-null mice) enhances the susceptibility of inflammation and atherosclerosis lesions caused by a high-fat diet [62]. Further research found that global β3 deletion or transplantation of β3^−/−^ bone marrow caused atherosclerotic lesions to aggravate due to the increase in the number of smooth muscle cells (SMCs) recruited into the plaque. The authors believe that macrophage β3 deletion plays a leading role in this progress [63]. However, there are reports contrary to the above studies. The adhesion of platelets to ECs depends on the interaction of soluble fibrinogen and αIIbβ3. Platelets lacking αIIbβ3 cannot firmly adhere to activated ECs [64]. In a study of primary stroke, it was shown that the area of cerebral infarction in mice lacking GPIIb compared to wild-type mice significantly decreased. Similarly, the atherosclerotic lesions in the carotid artery and aortic arch in ApoE^−/−^ mice lacking GPIIb were significantly reduced or even absent. The authors suggest that this may be due to the decrease in the number of recruited platelets adhering to active ECs [65]. The degree of atherosclerotic lesions in GPIIb or GPIIIa deficiency in atherosclerosis-prone mice may be related to the vitronectin receptor αVβ3. After all, αIIbβ3 is only expressed by megakaryocytes and platelets, while αVβ3 is expressed on both vascular cells and monocytes. αvβ3 also plays a vital role in promoting platelet adhesion. Inhibiting αvβ3 can suppress inflammation and smooth muscle recruitment, thereby weakening atherosclerosis [66]. In order to clarify the specific role of αIIbβ3 in AS, it is necessary to use platelet-specific knockout β3 for in-depth research.

### 3.2. GPIb-V-IX Complex

One of the important adhesion receptors on platelets is glycoprotein (GP) Ib. After vascular injury, collagen fibers are exposed, and the GPIb-V-IX complex binds to vWF and is fixed on the collagen fibers. This process promotes platelet actin aggregation and skeletal reorganization on the platelet surface, and mediates platelet activation and adhesion to ECs [67]. The number of platelets was significantly decreased, and plaque formation was dramatically limited in ApoE^−/−^ mice after GPIb antibody treatment [68]. Consistent with this, atherosclerotic lesions were reduced after GPIbα-specific knockout of bone marrow cells in Ldlr^−/−^ mice. However, only removing the extracellular domain of GPIbα that interacts with various ligands (vWF, P-selectin, Mac-1) can reverse this phenotype [69]. This result seems to indicate that the interaction of the extracellular domain of GPIbα on platelets with leukocytes or ECs may be redundant during plaque formation. However, this is contrary to previous research, which shows that vWF or P-selectin deficiency can reduce atherosclerotic lesions [70,71], suggesting that VWF and P-selectin may also affect atherosclerosis by binding to other receptors instead of GPIb. Low-fat feeding on GPIb^−/−^/ApoE^−/−^ mice showed no decrease in atherosclerotic plaque size and composition, and even an increasing trend [72]. This is contrary to the above GPIbα study, which may be caused by the different feed formula, the changes in the number and the state of platelets in mice after gene deletion, the release of platelet particles, and the number of inflammatory cells (SMCs, monocytes).

### 3.3. GPVI

The platelet-specific immunoglobulin superfamily receptor GPVI can induce platelet activation, adhesion, and aggregation, when activated by collagen or fibrinogen [73,74]. Platelet aggregation may have certain effects on atherosclerotic vessel wall homeostasis and atherosclerotic plaque rupture. Previous studies have found GPVI expression in the core region of human plaques [75]. Clinical studies have found that in patients with acute coronary syndrome (ACS) and unclear electrocardiogram, soluble dimeric GPVI (sGPVI) levels in coronary atherosclerotic heart disease (CAD) patients are generally elevated [76,77]. The sGPVI receptor has high affinity with immobilized collagen within atherosclerotic plaques [78]. In vitro animal experiments found that inhibition of GPVI by soluble GPVI-Fc or anti-GPVI antibodies protects atherosclerosis in cholesterol-fed rabbits and ApoE^−/−^ mice [79]. This may be due to GPVI blocking partially inhibiting the adhesion of platelet and fibronectin to activated ECs. Another report showed that cross-linking GPVI-Fc enhances the inhibition of human plaque- and collagen-induced platelet aggregation, further supporting this [80]. Moreover, the use of microbubble-targeted contrast-enhanced ultrasound (CEU) combined with soluble GPVI can effectively localize and diagnose atherosclerotic lesions in ApoE^−/−^ mice [81]. Subsequent studies also showed that GPIb-Fc had a stronger antithrombotic effect on the rupture site of high-risk lesions of atherosclerosis [82]. Thus, GPVI-FC may be a potential therapeutic target for plaque rupture or endothelial injury. The current research on platelet GPVI mainly focus on the damage of blood vessel integrity and thrombosis caused by platelet activation or platelet and endothelial adhesion. Although clinical studies have also shown that GPVI can be used as a marker of atherosclerotic plaque, the relative contributions of platelet GPVI in vivo contributing to atherosclerosis are still incompletely understood. The knowledge of its molecular mechanism is very limited.

### 3.4. P-Selectin

P-selectin is encoded by the selp gene, stored in α-granules of platelet and Weibel Palace bodies of endothelial cells [83]. P-selectin mediates the rolling of leukocytes on activated endothelium, the first step in the cell adhesion cascade [84,85]. P-selectin is not only a marker of platelet activation but also an important signaling molecule and mediator of cell–cell interactions, which seems to be very important for platelet-induced inflammation models and subsequent atherosclerosis. Destruction of the mouse p-selectin gene can significantly inhibit leukocyte rolling and delay the recruitment of monocytes to sites of inflammation [86]. Similarly, in Ldl-r^−/−^ mice or ApoE^−/−^ mice, the lack of P-selectin significantly reduced the fatty streak lesion size and macrophage infiltration [87,88]. Further studies have shown that both endothelial P-selectin deficiency and platelet P-selectin deficiency can reduce atherosclerosis lesions and macrophage infiltration in ApoE^−/−^ mice with high-fat feeding. The latest research also reports that endothelial cell-derived P-selectin binds to p-selectin glycoprotein ligand 1 (PSGL-1) on dendritic cells (DCs), promotes an inflammatory response through the TLR4 signaling pathway, and participates in the progression of atherosclerosis [89]. In addition, platelet P-selectin can induce platelet adhesion to monocytes via the P-selectin-PSGL-1 axis, which further evokes the expression of some inflammatory factors (TNF-α, MCP-1) [90,91]. Increased binding of platelet-monocytes is also considered as a marker of acute coronary syndrome [92]. The expression of inflammatory factors accelerates the transformation of monocytes to macrophages and promotes atherosclerosis. Furthermore, Zhang et al. used mice expressing human SELP transgenes to further verify the atherosclerotic effect of P-selectin [93]. These combined data support an important role for P-selectin in murine or human atherogenesis. Treatment targeting the pathological expression or function of human p-selectin may effectively improve the prevention of atherosclerosis-related cardiovascular disease.

### 3.5. Thrombin Receptors

Thrombin is a multifunctional serine protease produced at the site of vascular injury, which converts fibrinogen to fibrin, activates platelets, and causes multiple actions of various cell types. The cellular effects of thrombin are mediated in part by the G protein-coupled receptor family protease-activated receptors (PARs, include PAR1~4) [94]. The role of PARs in the progression of atherosclerosis has been confirmed [95,96,97,98,99], but the link between platelet-derived PARs and the actual pathogenesis of AS is less clear. Interestingly, while human platelets utilize two thrombin receptors, PAR-1 and PAR-4, mouse platelets utilize PAR-3 and PAR-4 [100]. PAR1 and PAR3 mediate the activation of platelets at low thrombin concentrations and PAR4 can mediate platelet activation but only at high thrombin concentrations [100]. This may also be the reason why PAR1-inhibited human platelets are analogous to PAR3-deficient mouse platelets, as both rely on PAR4 for thrombin signaling [101,102]. However, PAR3 is rarely studied mainly because of the extremely low expression in human platelets [101]. The expression of PAR1 is abnormally elevated in human atherosclerotic arteries [103]. Application of PAR1 pepducin PZ-128 can inhibit macrophage infiltration of atherosclerotic lesions in mice [95,104]. The latest research report further confirms the role of PAR1 in the development of diet-mediated atherosclerosis, as one of the articles believes that PAR1 can promote AS in ApoE^−/−^ mice by inhibiting the efflux of cholesterol in macrophages and smooth muscle cells and the recruitment of leukocytes [96]. Another article reported that thrombin can trigger the formation of foam cells by inducing the expression of CD36, and this process relies on the activation of the PKCθ signaling pathway mediated by PAR1 [105]. PAR4 has evolved a unique strategy for interacting with thrombin in human and mouse platelets. A study with PAR4 deficiency in ApoE^−/−^ mice showed no significant difference in the plaque area compared with PAR4^+/+^/ApoE^−/−^ mice [106]. This indicates that PAR-4 is not required for the development of early atherosclerotic lesions. More recently, long-term administration of the thrombin inhibitor dabigatran in diabetic rats can cause upregulation of vascular PAR4, leading to increased platelet aggregation and coronary lipid deposition [107]. Therefore, the role of PAR4 in atherogenesis is still controversial, and platelet-derived PAR4 needs further study through platelet-specific knockout mice.

### 3.6. ADP Receptors

Along with thrombin, ADP has been recognized an important primary platelet agonist. ADP can be actively secreted by platelet-dense granules. Human platelets include three different ADP receptors: P2Y1, P2Y_12_, and P2X1. P2X1 is a ligand-gated ion channel, and P2Y1 and P2Y_12_ are receptors linked to two different G proteins. In platelets, the P2X1 receptor triggers transient shape change and participates in collagen- and shear-induced platelet aggregation, and may contribute to thrombus formation in the context of inflammation [108,109]. P2X1^−/−^ mice displayed resistance to the systemic thromboembolism induced by injection of a mixture of collagen and adrenaline and also to the thrombosis triggered by a laser [108]. Overall, combined with current research, it is believed that P2X1 receptors are closely related to thrombosis and do not play a role in early atherosclerosis. Platelet P2Y1-related research mostly focuses on thrombosis and vascular inflammation. Activation of P2Y1 initiates shape change in platelet and leads to ADP-induced platelet aggregation [110,111]. The P2Y1 receptor antagonist MRS2500 can effectively inhibit experimental thrombosis in vivo [112]. In addition, the P2Y1 receptor can also facilitate the interaction between platelets and leukocytes resulting in leukocyte activation [113], which may be related to the platelet-dependent recruitment of leukocytes to lung tissue in the case of allergic airway inflammation [114]. However, this does not exclude the role of P2Y1 derived from endothelial cells or leukocytes. Indeed, the endothelial cell P2Y1 receptor strongly contributes to leukocyte recruitment and exposure of adhesion molecules (P-selectin, ICAM-1, and VCAM-1) during inflammation [115], and P2Y1^−/−^/ApoE^−/−^ mice showed reduced macrophage infiltration and atherosclerosis lesions [116]. The reason why platelet P2Y1 has a weak role in atherosclerosis or inflammation may be related to its expression on platelets. After all, approximately 150 copy numbers of P2Y1 receptors are expressed on each platelet, which is very low [117]. P2Y_12_ is currently a clinically determined antiplatelet drug target, and P2Y_12_ antagonists, such as clopidogrel, prasugrel, or ticagrelor, are clinically used in patients with coronary heart disease to inhibit platelet aggregation [118]. Moreover, animal studies have also shown that the P2Y_12_ receptor may participate in atherogenesis by promoting the proliferation and migration of VSMCs, and endothelial dysfunction [119,120,121]. Deletion of the P2Y_12_ gene in ApoE^−/−^ mice or LDL^−/−^ mice resulted in shrinking of the lesion area, decreased monocyte/macrophage infiltration, and increased fiber in plaque after a high-fat diet for 12 weeks [119,122]. Bone marrow transplantation experiments further demonstrate that platelet-expressed P2Y_12_ is a key factor leading to atherosclerosis, but the role of smooth muscle cell P2Y12 is not excluded [119]. Subsequently, a study by West et al. [120] showed that the lack of P2Y_12_ derived from the blood vessel wall rather than platelet can reduce the lesion size of atherosclerotic plaques in ApoE^−/−^ mice. In summary, these results indicate that the early occurrence of atherosclerosis is mainly related to the expression of P2Y_12_ in the vessel wall not in the platelet.

## 4. Platelet-Derived Proinflammatory Mediators in Atherogenesis

After platelet activation, a large amount of adhesive and proinflammatory substances stored in α granules and dense tubular systems are released. These substances interact with circulating leukocytes and blood vessel wall cells to induce a strong inflammatory response [1]. Proinflammatory substances released by platelets usually include various cytokines (IL1β, CD40L), chemokines (CCL5, CXCL4, CXCL7, CXCL12), growth factors (PDGF), and damage-related molecular model molecules DAMPs (HMGB, cyclophilin A) (Figure 3) [68,123]. The complex functional relationships of different platelet-derived mediators provide a mechanistic framework for the insight into the mechanisms by which platelets promote atherosclerosis.

### 4.1. Cytokines

IL-1β and CD40 ligand (CD40L, CD154) are important proinflammatory cytokines. IL-1β is synthesized and released after platelet activation and has become a target for coronary heart disease treatment. Activated platelet-synthesized interleukin (IL)-1β can induce the inflammatory response of ECs and promote the adhesion of platelets to leukocytes under the synergistic effect of e-selectin, ICAM-1, and chemokines [123]. In patients prone to development of atherosclerosis, polymorphism of the IL-1β gene cluster is associated with the extent of coronary arteries lesions, especially IL1B: −511 and −31 C/T polymorphism [124,125]. CD40L is stored in the cytoplasm of resting platelets, and is rapidly released to the membrane surface after platelet activation, and then it is cleaved to form a soluble functional fragment, sCD40L [1]. Platelets are the main source of circulating sCD40L, and elevated levels of circulating sCD40L have been reported in patients with hypercholesterolemia and diabetes [126,127,128]. This suggests that platelet-derived sCD40L can be an indicator for predicting postoperative risk of cardiovascular disease. IL-1β and CD40L on platelets induces ECs to express adhesion molecules (ICAM-1/VCAM-1) and secrete inflammatory factors (IL-8 and MCP-1), and promote the recruitment and extravasation of leukocytes at the site of injury, directly triggering an inflammatory response in the vessel wall [129]. Interrupting CD40 signal transduction with CD40L antibodies or CD40L deficiency in ApoE^−/−^ mice can improve atherosclerotic lesions, reducing the infiltration of intermediate macrophages and T lymphocytes [130,131]. In clinical studies, the upregulation of platelet CD40L indicates a poor prognosis in stroke patients and is associated with increased platelet-mononuclear aggregate formation [132]. These results indicate that CD40L plays an important role in AS. However, Bavendiek et al. [133] showed that a CD40L deficiency on bone marrow-derived cells does not alter diet-induced atherosclerosis in hypercholesterolemia mice. This suggests that CD40L mainly regulates the occurrence of atherosclerosis through its expression on non-hematopoietic cell types, and platelet CD40L may not participate in AS. Moreover, the lack of CD40L will affect the stability of arterial thrombosis and delay arterial occlusion in vivo [134]. Therefore, simply blocking CD40L may not be feasible in the clinical treatment of atherosclerosis and other cardiovascular diseases, because long-term inhibition will increase the risk of thromboembolic events. Conceivably, more targeted intervention strategies in CD40 signaling will have less deleterious side effects.

Meanwhile, platelets also express substantial levels of CD40, which is the alleged counter receptor for CD40L [135]. CD40 is different from CD40L, and the role of the receptor CD40 in the development of atherosclerosis remains disputable. Zirlik [136] reported that CD40 deficiency in Ldlr^−/−^ mice does not ameliorate atherosclerosis, although the endothelial CD40-TNF receptor-related factor (TRAF) signaling pathways have been proven to promote atherosclerosis. Lutgens [137] also reported that deficiency in hematopoietic CD40 in Ldlr^−/−^ mice or genetic interruption of CD40-TRAF6 signaling in ApoE^−/−^ mice reduces atherosclerosis and increases plaque fibrosis. Gerdes [138] further reported that platelet CD40 promotes atherogenesis by stimulating endothelial cell activation and recruiting leukocytes. In summary, the current findings suggest that platelet CD40 and CD40L can serve as a key interface between inflammation, thrombosis, and atherosclerosis and are attractive potential therapeutic targets for cardiovascular disease.

### 4.2. Chemokines

#### 4.2.1. CXCL4

Platelet factor 4 (PF-4 or CXCL4), a member of the C-X-C subfamily of chemokines, is stored in the α-particles of platelet, and is extremely abundant in platelets. It is quickly mobilized and released into plasma when platelets are activated, and is the first type of medium for early thrombosis or plaque formation [139]. The presence of PF4 in atherosclerotic lesions correlates with clinical parameters in patients with atherosclerosis [140]. Additionally, co-localization of PF4 and ox-LDL can be observed in human atherosclerotic lesions, especially in macrophage-derived foam cells [139,141]. In vitro experiments found that PF4 not only induces the differentiation of monocytes into macrophages, named “M4” [142,143], but also promoted the binding of ox-LDL to vascular cells and macrophages and the accumulation of cholesterol esters [140]. These observations suggest that platelet activation may promote the accumulation of harmful lipoproteins and thus promote atherosclerosis. The more direct evidence is that PF4^−/−^; ApoE^−/−^ mice have a strong decrease in atherosclerotic lesion formation compared to ApoE^−/−^ mice [144]. The reason for the reduction in plaque is most likely that PF4 inhibits the expression of the hemoglobin scavenger receptor (CD163) in proinflammatory macrophages, and CD163 has anti-lipid peroxidation and anti-inflammatory effects [145]. Normally, PF4 is more likely to form heterodimers, dimers, and oligomers with CCL5, inducing the binding of monocytes to ECs, thereby promoting the transmigration of monocyte into the subendothelial space.

#### 4.2.2. CCL5

CCL5 (RANTES) is the most expressed chemokine during platelet transcription, and mainly activates CCR5 and CCR1 receptors. The microparticles released after platelet activation promotes the delivery of RANTES to the surface of monocytes in atherosclerotic arteries [146]. RANTES deposition enhances the recruitment of monocytes by inflammatory microvascular or aortic ECs, a process that depends on P-selectin expression [147,148]. Treatment with the RANTES antagonist Met-RANTES in Ldlr^−/−^ mice reduces leukocyte infiltration and diet-induced atherosclerosis [149,150]. Clinical studies in patients with acute myocardial infarction and stable angina pectoris indicate that RANTES is more likely to be a biomarker for the presence of chronic coronary artery disease and is critical for the initial stage of atherosclerotic plaque formation [151]. Although RANTEs can be used as an indicator of AS inflammation, the specific role of platelet-derived RANTEs in AS needs more evidence. In fact, platelet-derived RANTES, in most cases, together with other mediators, induce monocyte inflammatory factor secretion (McP-1, McP-4, and IL-8) and accelerate AS [3]. One of the most common combinations is that CXCL4 and CCL5 interact to form heterodimers, then synergistically recruit monocytes to inflammatory vascular ECs [146] and induce the release of neutrophil extracellular traps (NETs) [152]. Disrupting the synergistic effect of CXCL4/CCL5 with peptide inhibitors MKEY will reduce leukocyte recruitment, NETosis formation, and ultimately reduce infarction size [153,154,155]. In conclusion, compared with CCL5 deficiency [156], blocking chemokine heterodimers can reduce inflammatory side effects and maintain normal immune defense.

#### 4.2.3. CXCL7

CXCL7 is abundant in platelets; it is divided into several variants by pre-platelet basic protein (pre-PBP), including platelet basic protein (PBP), connective tissue-activating peptide III (CTAP-III), β thrombin (β-TG), and neutrophil-activating peptide 2 (NAP-2) [157]. NAP-2 is considered to be the only variant with chemotactic activity. Platelet-derived NAP-2 (CXCL7) deficiency or blockade of its receptor CXCR1/2 can significantly reduce thrombosis-induced neutrophil migration [158]. Patients with acute myocardial infarction were treated with PCI and found that plasma CXCL7 levels were negatively correlated with myocardial dysfunction [159]. However, there is no more relevant research reported in cardiovascular, especially atherosclerosis, patients. To the best of our knowledge, there are few studies on CXCL7 in the progress of AS. Additionally, there are few reports related to mononuclear/macrophages either. Further research is still needed on the role of platelet-derived CXCL7 in atherosclerosis.

#### 4.2.4. CXCL12

The chemokine CXCL12, also known as stromal cell-derived factor-1 (SDF-1), is stored in platelet α granules. Early research found that high expression of CXCL12 was detected in SMCs, ECs, and macrophages of human atherosclerotic plaques from human carotid arteries [160]. However, recent studies by Merckelbach et al. [161] on human carotid atherosclerotic plaques have shown that CXCL12 is expressed only on the macrodissected areas of macrophages. The CXCR12 receptor CXCR4 is expressed on plaque macrophages, SMCs, and leukocytes. The reason for this difference may be related to the disease status and complications of patients with different plaque samples. In addition, the lower sensitivity of IHC staining and the shorter half-life of CXCL12 may also lead to poor observation of CXCL12 staining. In vitro experimental studies have found that platelet-derived CXCL12 can participate in the regulation of monocyte function, and induce the differentiation of monocytes into macrophages and foam cells through the receptors CXCR4-CXCR7 [162]. Therefore, CXCL12 may participate in AS through mononuclear differentiation. Animal studies further found that systemic treatment of CXCL12 in ApoE^−/−^ mice promoted the mobilization and accumulation of smooth muscle progenitor cells at the site of vascular injury, thereby increasing plaque stability [163]. However, there is no evidence to show whether platelet-derived CXCL12 is playing a role here, thus further studies are necessary to elucidate the exact role of the platelet-derived CXCL12 in atherosclerotic plaques and potential impact on plaque vulnerability.

### 4.3. Other Platelet-Derived Inflammatory Mediators

The number of inflammatory mediators released from activated platelets is rapidly increasing. Platelet-derived growth factor (PDGF) is stored and released by α granules from activated platelets but can be widely secreted by macrophages, VSMCs, and endothelial cells [164]. The presence of PDGF was detected in the atherosclerotic vessel wall, especially PDGF-A and PDGF-B [165]. Simultaneously, PDGF receptors (PDGFR), PDGFR-α and PDGFR-β, expressed by macrophages and SMCs, are also significantly upregulated [166]. Multiple studies have shown that PDGF and PDGFR mainly regulate the development of atherosclerotic lesions by inducing the migration and proliferation of SMCs [167,168,169]. In ApoE^−/−^ mice, blocking the PDGF-PDGFR-β pathway by neutralizing antibodies or chemical inhibitors prevented vascular smooth muscle cell accumulation and delayed fibrous cap formation [170,171]. In addition, enhancement of PDGF signaling led to severer inflammation, and promoted the progression of atherosclerosis in ApoE^−/−^ mice [172]. However, at present, all studies have not pointed out how PDGF derived from platelets affects the expression of PDGFR in ECs, VSMCs, and infiltrating macrophages participating in vascular inflammation and remodeling.

Amphoterin (HMG1) is an endogenous protein in human platelets, which is exposed on the surface during platelet activation [173]. HMGB1 expression in human atherosclerotic plaques and coronary artery thrombi was identified. Therapeutic blockade of HMGB1 reduced the development of diet-induced atherosclerosis in ApoE^−/−^ mice [174]. Furthermore, platelet-derived HMGB1 induces NETs formation and thrombosis [175,176]. This suggests that platelet-derived HMGB1 may be involved in the progression of both atherosclerosis and atherosclerotic thrombosis. Cyclophilin A (CyPA), a protein released when platelets are activated, is found in atherosclerotic plaques [177]. CyPA stimulates migration and proliferation of VSMCs, expression of adhesion molecules in endothelial cells, and inflammatory cell chemotaxis, and promotes atherosclerosis in ApoE^−/−^ mice [178,179]. In addition, platelets also release several other proteins that may be related to atherosclerosis, such as granule protein III, histamine, etc. At present, we have less research on these “new” platelet-derived mediators, and future research should be an attempt to determine the role of these inflammatory factors in the formation of plaque.

## 5. Platelet Migration in Atherogenesis

Platelets guard blood vessels as a “patrol” to maintain vascular homeostasis in the physiological environment. Upon encountering endothelial injury or inflammation, the circulating platelets are recruited to the damaged sites of inflammation by various inflammatory factors and chemokines. Meanwhile, the platelets activated in the circulation can also release inflammatory factors and chemokines to promote the combination of platelets and immune cells. The interaction between these molecules and cells has been widely reported in atherosclerosis. However, the direct interaction of platelets with different subpopulations of leukocytes in atherosclerotic plaques remains unclear. Barrett et al. [180] found PF4 expression in a subset of plaque macrophages based on their single-cell RNA sequencing in hypercholesterolemic mice, thus they suggested increased macrophage-platelet aggregates accumulate in atherosclerotic plaques. However, PF4 can also be expressed by a small number of macrophages [140,181], so whether platelets are present in atherosclerotic plaques needs further evidence for this to be confirmed. This may be proved by directly observing platelet markers (e.g., CD41, CD42d) at different stages of plaque formation by immunofluorescence staining. The study published by Gaertner et al. [182] identified that platelets have the capacity for locomotion at the level of cell biology for the first time, and they can migrate autonomously at the site of vascular injury. Platelets that migrate at the site of infection can help trap bacteria and promote the activation of NETs. In addition, platelets can penetrate into the lung tissue of allergic asthma through the reaction to allergens [183]. All these data indicate that the migrated platelets may promote tissue damage as inflammatory cells. In the process of atherosclerosis, whether platelets can actively migrate into the plaque, and how the migrated platelets act on the plaque formation needs in-depth study. Kraemer et al. [184] used transwell experiments in vitro found that CXCL12 can stimulate the migration of platelets on the surface of fibrinogen and collagen. The migration of platelets did not depend on either GPVI or GPIb/GPIIbIIIa. Receptors on the platelet surface recognizing and participating in the process of platelet migration need further exploration. Interestingly, Witte et al. [185] used platelet CXCR4-specific knockout mice, and for the first time showed that platelet-derived CXCL14 could induce monocyte migration through CXCR4 and enhance monocyte phagocytosis [186]. Controversially, some early in vitro flow chamber experiments believed that the platelets were still retained on the top of endothelial surface after they promoted monocyte transmigration [187,188]. Conclusively, whether platelets autonomously transmigrate, or could be carried by leukocytes into the subendothelium and play a role in atherogenesis still requires further proof. This could be addressed in in vitro transmigration models using cultured endothelial cells, or an in vivo animal model, combined with a pro-atherosclerotic factor, such as pathological blood flow.

## 6. Concluding Remarks and Future Perspectives

Platelets are important blood cells in the body. The initial physiological functions of platelets are mainly to stop the bleeding and keep the vascular endothelium intact. However, now, accumulating evidence indicates that platelets, as a novel immune and inflammatory cell, can modulate the inflammatory response of neighboring cells, such as leukocytes, ECs, and vascular SMCs. Simultaneously, platelets also receive the inflammatory signals produced by their neighboring cells. This double effect between platelets and neighboring cells plays roles in both the early and late stage during atherosclerosis. Although a variety of antiplatelet drugs have been widely used in patients with atherosclerotic diseases, platelet-mediated inflammation (complications due to side effects of the drug) appears to be operating. Therefore, the latest advances in understanding how platelets participate in the formation of atherosclerotic plaques will provide important clues for antiplatelet drugs in the prevention and treatment of cardiovascular diseases, especially atherosclerosis. Several issues need to be addressed in future studies: (1) whether and how platelets regulate atherosclerotic plaque formation under disturbance flow; (2) how platelets foster monocyte recruitment to atherosclerotic lesions and how they reprogram macrophages; (3) the association of platelet-derived inflammation mediators with atherogenesis should be explored in detail, such as using platelet-specific knockout mice; (4) further study is required to confirm whether platelet heterogeneity has been involved by using single-cell sequencing technology; and (5) whether platelets actively migrate to the subintima to participate in atherogenesis.

## Figures and Tables

**Figure 1 ijms-21-09760-f001:**
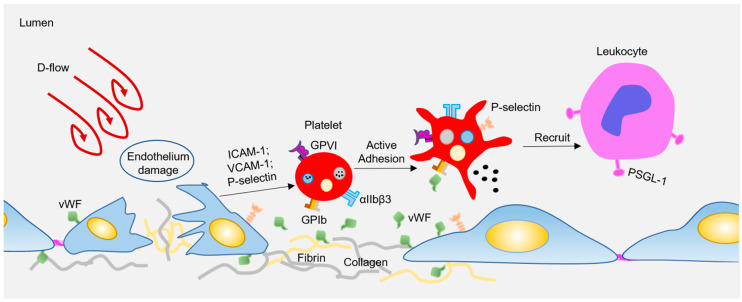
Disturbed flow-regulated platelets participate in the plaque formation. Disturbed flow activates endothelial cells (ECs), resulting in elevated expression of adhesion molecules and deposition of adhesion proteins. All these adhesion molecules and proteins interact with platelets via surface receptors, leading to platelet activation. Activated platelets recruit circulating leukocytes by P-selectin or other releasing inflammatory factors, therefore participating atherogenesis.

**Figure 2 ijms-21-09760-f002:**
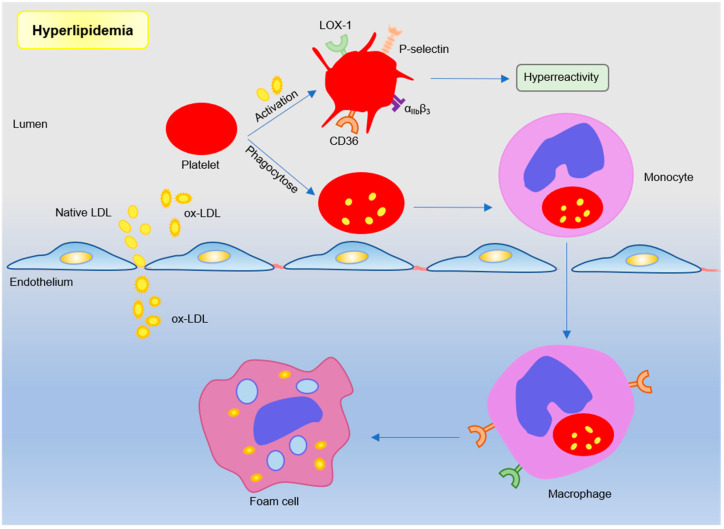
Platelets participate in atherosclerosis under hyperlipidemic conditions. Lipoproteins, such as low-density lipoproteins in plasma, trigger platelet activation by binding sites. Or monocytes engulf low-density lipoprotein (LDL)-containing apoptotic platelets and migrate under the inner membrane to differentiate into macrophages and foam cells. Subendothelial oxidized LDL further participates in atherosclerotic plaque formation by binding to scavenger receptor CD36 and LOX-1.

**Figure 3 ijms-21-09760-f003:**
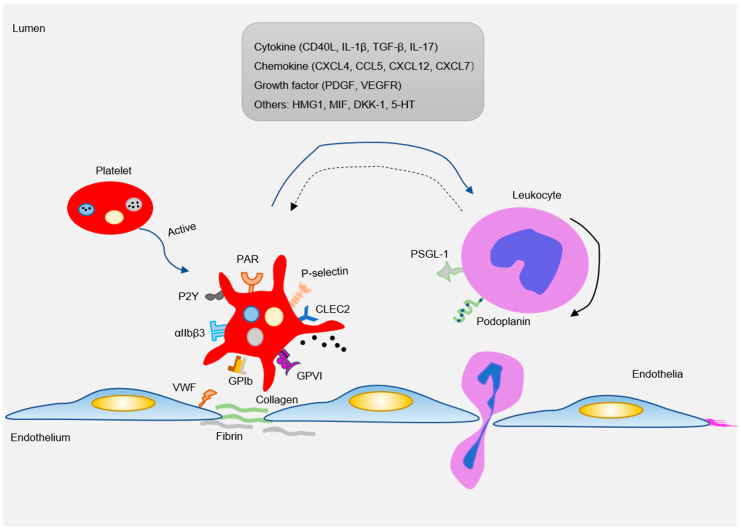
As an inflammatory mediator, platelets interact with ECs and leukocytes (mainly monocytes) to participate in the formation of atherosclerotic lesions. Pro-atherosclerosis factors can easily lead to platelet activation. The activated platelets express a variety of adhesion receptors. These receptors bind their matrix proteins (GPIb-vWF, GPVI-collagen, and GPIIb-IIIa-fibrinogen) to mediate platelet adhesion on ECs or leukocytes (P-selectin-PSGL-1 and Clec2-PDPN). Moreover, platelet activation releases a variety of inflammatory factors (cytokines, chemokines, growth factors, and others). These inflammatory mediators can also be induced and expressed in neighboring cells, such as monocytes/macrophages, neutrophils, and ECs, which in turn affected platelets. Blue arrow: the factor released by platelets interact with leukocytes; Black solid arrow: the rolling of leukocytes; Blue dotted arrow: activated leukocytes in turn affect platelets.

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
