# Peer review of "Targeting Platelet in Atherosclerosis Plaque Formation: Current Knowledge and Future Perspectives"

_ijms, 2020, doi:10.3390/ijms21249760_

Round 1
Reviewer 1 Report
This is a very interesting review article, well focused on a topic of major interest, and of high medical utility, which addresses mechanisms and development of atherosclerosis, with the major involvement of platelets. The authors did a huge documentation analysis, and extracted very useful information for redacting this report. Congratulations for this deepful investigation on the atherosclerosis process and on the role of platelets and other blood cells.
The major concern that I can outpoint is that this article appears too much as a catalogue of scientific information on vessel wall, atherosclerosis development, platelets and their interactions with blood components, blood cells and endothelium/sub-endothelium, and it lacks synthetic presentation and it needs a link with medical aspects. Some attempts are done for presenting how all the described activities can act in synergy between them, and figures 1, 2 and 3 are very informative and useful. However, a too large part of the article describes platelet receptors, proteins, chemokines, growth factors and interactions with monocytes, endothelial cells, neutrophils, blood lipids; list which is not exhaustive. This is interesting and sound. However, the reader understands that all can contribute in different ways to atherosclerosis and disease progression, but not how these various functions interact between them. Actually, they are not independent functions, but all are regulated and involved in processes in a coordinated manner, with different contributions and activities of the various axis according to the stimulation trigger. It would be nice if the authors try to include all their described activities and functions in a dynamic model of atherosclerosis initiation, development, control and finally plaques disruption. If this can be linked to associated diseases, this would be a great bonus.
Concerning minor comments, the authors need to revise some English wordings and sentences which are not clearly written.
In addition, some information reported is not right and needs to be corrected. This is summarized here beow.
In abstract, sentence lines 16-17 is not clear and needs revision of English language.
Line 24: in atherosclerosis, is inflammation the cause or the consequence of the disease?
Line 28: English requires revision.
Line 34: platelets are first involved in innate immunity, as they have no adaptive response. They can be activated by some bacterial polysaccharides, polyphosphates, etc., and are part of the first line immune response along with contact phase, complement pathways, toll like receptors.
Line 38: the sentence subject is platelets; it should then be "they are activated" and "then they adhere"
Line 43: can induce (not "induces")
Line 41: English wording requires attention.
Lines 55-56: vWF is involved in platelets adhesion, and should be reminded here.
Legend to figure 1: English language requires attention; the complex should be GPIb-IX-V and not GPIb-IV-X.
Same comment line 69.
Line 79: "stimulate" should be better than "induce".
Line 84: "on" in place of "or" (before platelets).
Line 85: the term "formulate" is not appropriate; it should be "expose".
Line 97: the prognosis mentioned should be precised.
Lines 103-104: English language requires attention; the sentence needs to be formulated differently.
Line 113: what is meant by "platelet activation alone".
Lines 139-141 need to be reformulated, as they are not clear.
Line 155: receptors (in place of "receptor").
Line 164 needs reformulation.
Line 204: "the authors" rather than "the author" (as they are 2).
Line 230: indicating "vWF and others" is not accurate enough; it would be better to detail in the text what are the "others".
Line 258: this is "Weibel Palace bodies" not "Wabi-Palace".
Line 265: protect in place of protects.
Line 266: remove the "dot" after [73] and reformulate the sentence.
Line 270: sP-selectin in plasma does not exert procoagulant activity by itself but is a marker of hemostasis activation, and associated to hypercoagulability.
Line 301: English language requires attention.
Line 328: the reference is not appropriate. There is no reference N° 156, and Gerdes et Al. is not cited in the references list.
Lines 334-336: this information is wrong. Platelet Basic Protein is the Precursor of CTAP-III, then of NAP-2 and finally of beta-thromboglobulin (not of PF4). It is stored in platelet alfa-granules, not synthesized in those granules.
Lines 372-374: the progressive proteolysis of PBP, generates first CTAP-III, then NAP-2 and finally Beta-TG. Beta-TG is the final product.
Reference 133 is not "Pelisek et al." but "Merckelbach et Al, 2018". This needs attention.
Reviewer 2 Report
This review paper by Wang and Tang is a very well readable comprehensive Review about the actual literature on platelets and atherosclerosis.
I just feel, that one aspect of platelet-mediated atherogenesis was simply forgotten to be mentioned: platelet migration. Some recent work has shown, that activates platelets migrate through the endothelium into the subintima. I wouls like to suggest to add a few lines on this issue.
suggested literature on this issue:
Migrating Platelets Are Mechano-scavengers that Collect and Bundle Bacteria.
J Mol Med (Berl). 2010 Dec;88(12):1277-88. doi: 10.1007/s00109-010-0680-8. Epub 2010 Sep 18. PMID: 20852838 minor points/typing errors: line 97...associsted with a bad prognosis (or: correlates with prognosis) line 136 ....actived lox-1 ... should propably be ...activated.... line 170 ...furtherleads... should be further leads....Author Response
Please see the attachment

Reviewer 3 Report
This review by Lei Wang et al., is almost repetitive of already published article “Platelets as therapeutic targets to prevent atherosclerosis”, Atherosclerosis 307 (2020) 97–108.
Therefore this review is not adding any further information to the topic. It would be good if authors would cite the above article and discuss some other aspects in addition to that.
Round 2
Reviewer 3 Report
The present form is improved as per my suggestion. Still one minor concern is there given below.
1. The newly inserted reference no 12 is having incomplete information. Lack of vol no, Issue no and page number??